# Prophet Attention:
# Predicting Attention with Future Attention

**Fenglin Liu[1], Xuancheng Ren[2], Xian Wu[3], Shen Ge[3], Wei Fan[3], Yuexian Zou[1,4], Xu Sun[2,5]**
[1]ADSPLAB, School of ECE, Peking University
[2]MOE Key Laboratory of Computational Linguistics, School of EECS, Peking University
[3]Tencent, Beijing, China   [4]Peng Cheng Laboratory, Shenzhen, China
[5]Center for Data Science, Peking University
{fenglinliu98, renxc, zouyx, xusun}@pku.edu.cn
{kevinxwu, shenge, Davidwfan}@tencent.com

## Abstract

Recently, attention based models have been used extensively in many sequence-to-sequence learning systems. Especially for image captioning, the attention based models are expected to ground correct image regions with proper generated words. However, for each time step in the decoding process, the attention based models usually use the hidden state of the current input to attend to the image regions. Under this setting, these attention models have a "deviated focus" problem that they calculate the attention weights based on previous words instead of the one to be generated, impairing the performance of both grounding and captioning. In this paper, we propose the Prophet Attention, similar to the form of self-supervision. In the training stage, this module utilizes the future information to calculate the "ideal" attention weights towards image regions. These calculated "ideal" weights are further used to regularize the "deviated" attention. In this manner, image regions are grounded with the correct words. The proposed Prophet Attention can be easily incorporated into existing image captioning models to improve their performance of both grounding and captioning. The experiments on the Flickr30k Entities and the MSCOCO datasets show that the proposed Prophet Attention consistently outperforms baselines in both automatic metrics and human evaluations. It is worth noticing that we set new state-of-the-arts on the two benchmark datasets and achieve the 1st place on the leaderboard of the online MSCOCO benchmark in terms of the default ranking score, i.e., CIDEr-c40.

## 1   Introduction

The task of image captioning [7] aims to generate a textual description for an input image and has received extensive research interests. Recently, the attention-enhanced encoder-decoder framework [2, 17, 20, 29, 38, 54] have achieved great success in advancing the state-of-the-arts. Specifically, they use a Faster-RCNN [2, 45] to acquire region-based visual representations and an RNN [14, 18] to generate the coherent captions, where the attention model [3, 32, 49, 53] guides the decoding process by attending the hidden state to the image regions at each time step. Many sequence-to-sequence learning systems, including machine translation [3, 49] and text summarization [58], have proven the importance of the attention mechanism in generating meaningful sentences. Especially for image captioning, the attention model can ground the salient image regions to generate the next word in the sentence [2, 26, 32, 53].

Current attention model attends to image regions based on current hidden state [49, 53], which contains the information of past generated words. As a result, the attention model has to predict

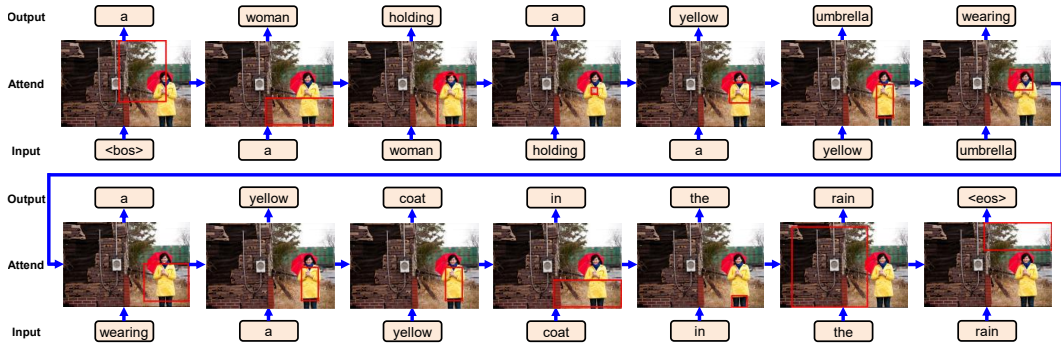

Figure 1: Illustration of the sequence of the attended image regions from a state-of-the-art system [20] in generating each word for a complete image description. At each time step, only the top-1 attended image region is shown [59]. As we can see, the attended image regions are grounded more on the *input* words than the *output* words, such as the timesteps that input *yellow* and *umbrella*, demonstrating poor grounding accuracies of the current attention model.

attention weights without knowing the word it should ground. Figure 1 illustrates a generated caption and the attended image regions from a state-of-the-art captioning system [20]. As we can see, the attended image regions are more grounded on current input word than the output one. For example, at the time step to generate the $5^{th}$ word *yellow*, the attended image region is the *woman* instead of the *umbrella*. As a result, the incorrect adjective *yellow* is generated rather than the correct adjective *red*. This is mainly due to the "focus" of the attention is "deviated" several steps backwards and the conditioned words are *woman* and *holding*; Another example is at the time step to generate the $7^{th}$ word *wearing*, the attended image region should be the *woman* instead of the *umbrella*. Although the generated word is correct, the unfavorable attended image region impairs the grounding performance [59] and ruins the model interpretability, because the attended image region often serves as a visual interpretation to qualitative measurement of the captioning model [9, 11, 33, 48, 59].

In this paper, to address the "deviated focus" issue of current attention models, we propose the novel Prophet Attention to ground the image regions with proper generated words in a manner similar to self-supervision. As shown in Figure 2, in the training stage, for each time step in the decoding process, we first employ the words that will be generated in the future, to calculate the "ideal" attention weights towards image regions. And then the calculated "ideal" attention weights are used to guide the attention calculation based on the input words that have already been generated (without future words to be generated). It indicates that the conventional attention model will be regularized by the calculated attention weights based on future words. We evaluate the proposed Prophet Attention on two benchmark image captioning datasets. According to both automatic metrics and human evaluations, the captioning models equipped with Prophet Attentions outperform baselines.

Overall, the contributions of this work are as follows:

- We propose Prophet Attention to enable attention models to correctly ground words that are to be generated to proper image regions. The Prophet Attention can be easily incorporated into existing models to improve their performance of both grounding and captioning.

- We evaluate Prophet Attention for image captioning on the Flickr30k Entities and the MSCOCO datasets. The captioning models equipped with the Prophet Attention significantly outperform the ones without it. Besides automatic metrics, we also conduct human evaluations to evaluate Prophet Attention from the user experience perspective. At the time of submission (2 June 2020), we achieve the 1st place on the leaderboard of the MSCOCO online server benchmark in terms of the default ranking score (CIDEr-c40).

- In addition to image captioning task, we also attempt to adapt Prophet Attention to other language generation tasks. We obtain positive experimental results on paraphrase generation and video captioning tasks.

The rest of the paper is organized as follows. Section 2 introduces the proposed Prophet Attention. Section 3 and Section 4 present the experimental results. Section 5 and Section 6 review the related work and conclude the paper, respectively.

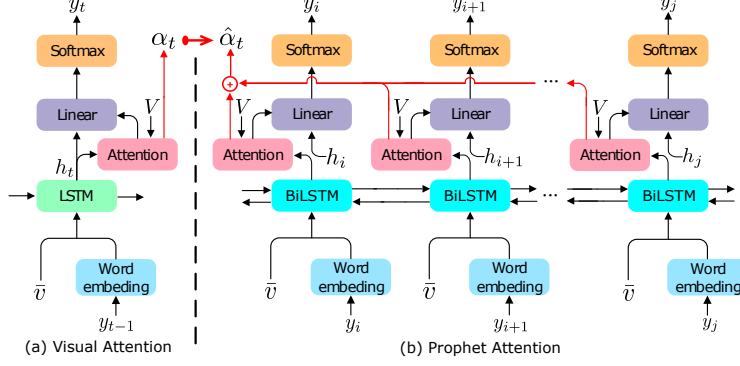

Figure 2: Illustration of the conventional attention model (left) and our Prophet Attention (right) approach. As we can see, our approach calculates "ideal" attention weights $\hat{\alpha}_t$ based on future generated words $y_{i:j}$ ($j \geq t$) as a target for the attention model based on previous generated words.

## 2 Approach

We first briefly review the conventional attention-enhanced encoder-decoder framework in image captioning and then describe the proposed Prophet Attention in detail.

### 2.1 Background: Attention-Enhanced Encoder-Decoder Framework

The conventional attention-enhanced encoder-decoder framework [2, 20, 32] usually consists of a visual encoder and an attention-enhanced language decoder.

**Visual Encoder**    The visual encoder represents the image with a set of region-based visual feature vectors $V = \{\boldsymbol{v}_1, \boldsymbol{v}_2, \ldots, \boldsymbol{v}_N\} \in \mathbb{R}^{d \times N}$, where each feature vector represents a certain aspect of the image. The visual features serve as a guide for the language decoder to describe the salient information in the image. In implementation, the Faster-RCNN model [2, 45] is widely adopted as the region-based visual feature extractor, which has achieved great success in advancing the state-of-the arts [17, 20, 54, 55].

**Attention-Enhanced Caption Decoder**    The left sub-figure in Figure 2 shows the widely-used attention-enhanced LSTM decoder [20, 32]. For each decoding step $t$, the decoder takes the word embedding of current input word $y_{t-1}$, concatenated with the averaged visual features $\bar{v} = \frac{1}{k} \sum_{i=1}^{k} v_i$ as input to the LSTM:

$$h_t = \text{LSTM}\left(h_{t-1}, [W_e y_{t-1}; \bar{v}]\right), \tag{1}$$

where [;] denotes the concatenation operation and $W_e$ denotes the learnable word embedding parameters. Next, the output $h_t$ of the LSTM is used as a query to attend to the relevant image regions in the visual feature set $V$ and generates the attended visual features $c_t$:

$$\alpha_t = f_{\text{Att}}(h_t, V) = \text{softmax}\left(w_\alpha \tanh\left(W_h h_t \oplus W_V V\right)\right), \quad c_t = V \alpha_t^{\text{T}}, \tag{2}$$

where the $w_\alpha$, $W_h$ and $W_V$ are the learnable parameters. $\oplus$ denotes the matrix-vector addition, which is calculated by adding the vector to each column of the matrix. Finally, the $h_t$ and $c_t$ are passed to a linear layer to predict the next word:

$$y_t \sim p_t = \text{softmax}\left(W_p[h_t; c_t] + b_p\right), \tag{3}$$

where the $W_p$ and $b_p$ are the learnable parameters. It is worth noticing that some works [2, 33, 54, 55] also attempt to append one more LSTM layer to predict the word, please refer Anderson et al. [2] for details. Finally, given a target ground truth sequence $y_{1:T}^*$ and a captioning model with parameters $\theta$, the objective is to minimize the following cross entropy loss:

$$\mathcal{L}_{\text{CE}}(\theta) = -\sum_{t=1}^{T} \log\left(p_\theta\left(y_t^* | y_{1:t-1}^*\right)\right). \tag{4}$$

As we can see from Eq. (2), at each timestep $t$, the attention model relies on $h_t$, which contains the past information of generated caption words $y_{1:t-1}$, to calculate the attention weights $\alpha_t$. Such reliance on the past information makes the attended visual features be less grounded on the word to be generated in the current timestep, which impairs both the captioning and grounding performance.

## 2.2 Prophet Attention: Predicting Attention with Future Attention

**Formulation** To enable the attention model to "undeviatingly" ground the image regions with the word to be generated, we propose the Prophet Attention. Specifically, we first adopt the conventional encoder-decoder framework to generate the whole sentence $y_{1:T}$. Then, for each time step $t$, Prophet Attention takes the future information $y_{i:j}$ ($j \geq t$) as input to calculate the attention weights $\hat{\alpha}_t$, which are naturally grounded on the generated word. In implementation, as shown in the right sub-figure of Figure 2, we employ a Bidirectional LSTM (BiLSTM) to encode the $y_{1:T}$, thus the information of $y_{i:j}$ is first converted to $h'_{i:j}$, and then the attention weights are calculated by the following equation:

$$\hat{\alpha}_t = f_{\text{Prophet}}(h'_{i:j}, V) = \frac{1}{j-i+1} \sum_{k=i}^{j} f_{\text{Att}}(h'_k, V). \tag{5}$$

where the attention model in Eq. (2) and Eq. (5) share the same set of parameters. We propose to use L1 norm between the $\alpha_t$ and $\hat{\alpha}_t$ as a regularization loss in training, which can be defined as:

$$\mathcal{L}_{\text{Att}}(\theta) = \sum_{t=1}^{T} \|\alpha_t - \hat{\alpha}_t\|_1, \tag{6}$$

where $\|\cdot\|_1$ denotes the L1 norm. By minimizing the loss in Eq. (6), the attention model converges the "deviated" attention weights $\alpha_t$ calculated on previous generated words $y_{1:t-1}$ towards "ideal" attention weights $\hat{\alpha}_t$ calculated on future generated words $y_{i:j}$ ($j \geq t$).

Then, to train the Prophet Attention, we incorporate the $\hat{\alpha}_t$ into the conventional encoder-decoder framework to re-generate the target ground truth $y_{1:T}^*$, which is defined as:

$$\hat{c}_t = V\hat{\alpha}_t^{\text{T}}, \quad y_t \sim p_t = \text{softmax}\left(W_p[h_t; \hat{c}_t] + b_p\right), \quad \hat{\mathcal{L}}_{\text{CE}}(\theta) = -\sum_{t=1}^{T} \log\left(p_\theta\left(y_t^*|y_{1:t-1}^*\right)\right). \tag{7}$$

Combining the loss $\mathcal{L}_{\text{CE}}(\theta)$ in Eq. (4), the loss $\hat{\mathcal{L}}_{\text{CE}}(\theta)$ in Eq. (7) and the loss $\mathcal{L}_{\text{Att}}(\theta)$ in Eq. (6), the full training objective is defined as:

$$\mathcal{L}_{\text{Full}}(\theta) = \mathcal{L}_{\text{CE}}(\theta) + \hat{\mathcal{L}}_{\text{CE}}(\theta) + \lambda\mathcal{L}_{\text{Att}}(\theta), \tag{8}$$

where $\lambda$ is the hyperparameter that controls the regularization. During training, we first pre-train the captioning model with Eq. (4) for 25 epochs and then use Eq. (8) to train the full model. In this manner, we can initialize proper parameter weights for Prophet Attention. In the inference stage, since the future words are invisible to current time step in language generation tasks, we follow the same procedure of conventional attention model in caption decoding.

In the following paragraphs, we introduce two variants of Prophet Attention.

**Constant Prophet Attention (CPA)** Since the attention weight is mainly determined by the single word that is to be generated at the current time step $t$, the intuition is to set $i = j = t$. In this manner, the CPA only uses the word $y_t$ to be generated to calculate the attention weights $\hat{\alpha}_t$:

$$\hat{\alpha}_t = f_{\text{Prophet}}(h'_{i:j}, V) = f_{\text{Att}}(h'_t, V). \tag{9}$$

With Eq. (9), the CPA grounds the output word at current time step to the attended image regions. However, in image captioning, when the output word $y_t$ is an attribute word that can be used to describe multiple objects in the image, the attention model may produce confusing attended image regions. For example, when there are a *black shirt* and *black pants* in the image, if we only adopt the word *black* to calculate the attention weights, the Prophet Attention model will be confused which image region it should attend to, as there are more than one proper image regions, i.e., *shirt* and *pants*. In addition, when the $y_t$ is a non-visual word, e.g., *of* and *the*, there is no suitable visual information at all [21, 32], so we should also remove (i.e., mask) the Prophet Attention to prevent it from affecting the learning of the captioning model.

**Dynamic Prophet Attention (DPA)** To tackle the problem of the CPA, we enable the Prophet Attention to attend to the image regions conditioned dynamically on the information of future time steps. In particular, for a noun phrase, e.g., *a black shirt*, we should treat all the words in it as a whole phrase instead of the individual words. Thus, for our Dynamic Prophet Attention (DPA), if the current

output word $y_t$ belongs to a noun phrase (NP), the DPA will adopt all the words in the noun phrase to calculate the attention weights $\hat{\alpha}_t$. Then, when the word is a non-visual (NV) word, we will remove (mask) our Prophet Attention model, i.e., remove the loss $\hat{\mathcal{L}}_{\text{CE}}(\theta)$ in Eq. (7) and the loss $\mathcal{L}_{\text{Att}}(\theta)$ in Eq. (6). For the remaining words, following the CPA, we directly set $i = j = t$. Specifically, in image captioning, the remaining words usually are verbs, which are used as the relationship words in the captions to connect different noun phrases. In brief, the Dynamic Prophet Attention is defined as:

$$\hat{\alpha}_t = f_{\text{Prophet}}(h'_{i:j}, V) = \begin{cases} \frac{1}{n-m+1} \sum_{k=m}^{n} f_{\text{Att}}(h'_k, V) & \text{if } y_t \in \text{NP: } y_{m:n} \\ \text{MASK} & \text{if } y_t \in \text{NV: } \{y_{\text{NV}}\}, \\ f_{\text{Att}}(h'_t, V) & \text{otherwise} \end{cases} \qquad (10)$$

where $\{y_{\text{NV}}\}$ denotes the set of all NV words. Through our approach, the attention model can learn to ground each output word $y_t$ to image regions without the ground-truth of grounding annotation.

## 3 Experiments

In this section, we first describe the used datasets and the experiment settings. Then we evaluate the proposed Prophet Attention from two perspectives: 1) Captioning: whether the proposed approach generates more appropriate image caption; and 2) Grounding: whether the proposed approach attends to the correct image regions in generating the corresponding word.

### 3.1 Datasets, Metrics and Settings

**Datasets**   We use the Flickr30k Entities [42] and the MSCOCO [7] image captioning datasets for evaluation. They contain 31,783 images and 123,287 images, respectively. Each image in these two datasets is annotated with 5 sentences. In addition to textual captions, Flickr30k Entities [42] contains 275,755 bounding boxes from 31,783 images and each bounding box is associated with the corresponding phrases in the caption. MSCOCO does not have bounding boxes.

**Metrics**   To measure captioning performance, we adopt the captioning evaluation toolkit [7] to calculate the standard metrics: SPICE [1], CIDEr [50], ROUGE [24], METEOR [4] and BLEU [39], among them, SPICE, which is based on scene graph matching, and CIDEr, which is built upon on n-gram matching, are specifically designed to evaluate captioning systems and are more likely to be consistent with human judgment [1, 50]. To measure the grounding performance, we adopt the metrics $F1_{\text{all}}$ and $F1_{\text{loc}}$ [59] which evaluates based on two factors: whether the correct word is generated and whether the correct image region is grounded.

**Settings**   In implementation, we use spaCy library [19] for noun phrase tagging. We set the $\lambda = 0.1$, according to the average performance on the validation set. We experiment on three representative models Up-Down [2], GVD [59] and AoANet [20]. Since our approach aims to force the conventional attention model can learn to ground each output word to image regions and is augmentative to the existing models, we keep the inner structure of the baselines untouched and preserve the original settings. Our code is implemented in PyTorch [41]. All re-implementations and our experiments were ran on V100 GPUs. Following common practice [2, 20, 28, 54, 55], we further adopt CIDEr-based training objective using reinforcement training [47]. In particular, inspired by the Chen et al. [8], we further introduce a simple Prophet Knowledge Distillation (PKD) trick to distill the future knowledge for image captioning. The introduced PKD trick can be used to boost the performance during the reinforcement training. To conduct a fair comparison, both the baseline models and our method have been equipped with the PKD trick. Due to limited space, for detailed description of the PKD trick and the settings, please refer to our supplementary materials.

### 3.2 Captioning Performance

**Offline Evaluation**   To conduct a fair comparison, we acquire the results according to the widely-used Karpathy test split [22]. The MSCOCO validation and test set contain 5,000 images each, and the number is 1,000 images for Flickr30k Entities. As shown in Table 1, for two datasets, all baselines equipped with our approach receive performance gains over all metrics. More encouragingly, based on the GVD [59] and AoANet [20], which are the previous state-of-the-arts on Flickr30k Entities and MSCOCO datasets, respectively, our approach sets the new state-of-the-art performance on the two benchmark datasets, achieving 62.7 and 133.4 CIDEr score on Flickr30k Entities and MSCOCO respectively, demonstrating the effectiveness and the compatibility of the proposed approach.

Table 1: Performance of offline evaluations on the Flickr30k Entities and the MSCOCO image captioning datasets. DPA represents the Dynamic Prophet Attention. B-4, M, R-L, C and S are short for BLEU-4, METEOR, ROUGE-L, CIDEr and SPICE, respectively. $*$ and $\dagger$ denote our own implementation and statistically significant results (t-test with $p < 0.01$), respectively. $\ddagger$ denotes the results from papers published after we submit to NeurIPS 2020 (2 June 2020).

| Methods | Flickr30k Entities | | | | | | Methods | MSCOCO | | | | |
| | $F1_{all}$ | $F1_{loc}$ | B-4 | M | C | S | | B-4 | M | R-L | C | S |
| --- | --- | --- | --- | --- | --- | --- | --- | --- | --- | --- | --- | --- |
| NBT [33] | - | - | 27.1 | 21.7 | 57.5 | 15.6 | Up-Down [2] | 36.3 | 27.7 | 56.9 | 120.1 | 21.4 |
| Up-Down [2] | 4.53 | 13.0 | 27.3 | 21.7 | 56.6 | 16.0 | ORT [17] | 38.6 | 28.7 | 58.4 | 128.3 | 22.6 |
| GVD [59] | 3.88 | 11.7 | 26.9 | 22.1 | 60.1 | 16.1 | AoANet [20] | 38.9 | 29.2 | 58.8 | 129.8 | 22.4 |
| Cyclical [35]$\ddagger$ | 4.98 | 13.53 | 27.4 | 22.3 | 61.4 | 16.6 | X-Trans. [38]$\ddagger$ | 39.7 | 29.5 | 59.1 | 132.8 | 23.4 |
| Up-Down$*$ | 4.19 | 12.1 | 26.4 | 21.5 | 57.0 | 15.6 | Up-Down$*$ | 36.7 | 27.9 | 57.1 | 123.5 | 21.3 |
| w/ DPA | **5.45**$\dagger$ | **15.3**$\dagger$ | **27.2**$\dagger$ | **22.3**$\dagger$ | **60.8**$\dagger$ | **16.3**$\dagger$ | w/ DPA | **38.6**$\dagger$ | **29.1**$\dagger$ | **58.3**$\dagger$ | **129.0**$\dagger$ | **22.2**$\dagger$ |
| GVD$*$ | 3.97 | 11.8 | 26.6 | 22.1 | 59.9 | 16.3 | AoANet$*$ | 38.8 | 29.0 | 58.7 | 129.6 | 22.6 |
| w/ DPA | **4.79**$\dagger$ | **15.5**$\dagger$ | **27.6**$\dagger$ | **22.6**$\dagger$ | **62.7**$\dagger$ | **16.7**$\dagger$ | w/ DPA | **40.5**$\dagger$ | **29.6**$\dagger$ | **59.2**$\dagger$ | **133.4**$\dagger$ | **23.3**$\dagger$ |

Table 2: Highest ranking published image captioning results on the online MSCOCO test server. c5 and c40 mean comparing to 5 references and 40 references, respectively. $\ddagger$ is defined similarly to Table 1. We outperform previously published work on major evaluation metrics. At the time of submission (2 June 2020), we also outperformed all unpublished test server submissions in terms of CIDEr-c40, which is the default ranking score, and ranked the 1st.

| Methods | BLEU-1 | | BLEU-2 | | BLEU-3 | | BLEU-4 | | METEOR | | ROUGE-L | | CIDEr | |
| | c5 | c40 | c5 | c40 | c5 | c40 | c5 | c40 | c5 | c40 | c5 | c40 | c5 | c40 |
| --- | --- | --- | --- | --- | --- | --- | --- | --- | --- | --- | --- | --- | --- | --- |
| Up-Down [2] | 80.2 | 95.2 | 64.1 | 88.8 | 49.1 | 79.4 | 36.9 | 68.5 | 27.6 | 36.7 | 57.1 | 72.4 | 117.9 | 120.5 |
| GLIED [28] | 80.1 | 94.6 | 64.7 | 88.9 | 50.2 | 80.4 | 38.5 | 70.3 | 28.6 | 37.9 | 58.3 | 73.8 | 123.3 | 125.6 |
| SGAE [54] | 81.0 | 95.3 | 65.6 | 89.5 | 50.7 | 80.4 | 38.5 | 69.7 | 28.2 | 37.2 | 58.6 | 73.6 | 123.8 | 126.5 |
| GCN-LSTM [55] | - | - | 65.5 | 89.3 | 50.8 | 80.3 | 38.7 | 69.7 | 28.5 | 37.6 | 58.5 | 73.4 | 125.3 | 126.5 |
| AoANet [20] | 81.0 | 95.0 | 65.8 | 89.6 | 51.4 | 81.3 | 39.4 | 71.2 | 29.1 | 38.5 | 58.9 | 74.5 | 126.9 | 129.6 |
| $\mathcal{M}^2$ Trans. [10]$\ddagger$ | 81.6 | 96.0 | 66.4 | 90.8 | 51.8 | 82.7 | 39.7 | 72.8 | 29.4 | 39.0 | 59.2 | 74.8 | 129.3 | 132.1 |
| X-Trans. [38]$\ddagger$ | **81.9** | 95.7 | **66.9** | 90.5 | **52.4** | 82.5 | **40.3** | 72.4 | **29.6** | 39.2 | **59.5** | 75.0 | **131.1** | 133.5 |
| Ours | 81.8 | **96.3** | 66.5 | **91.2** | 51.9 | **83.2** | 39.8 | **73.3** | **29.6** | **39.3** | 59.4 | **75.1** | 130.4 | **133.7** |

**Online Evaluation**    Following common practice [2, 20, 28, 54], we also submit an ensemble of four "AoANet w/ Dynamic Prophet Attention" models to online MSCOCO evaluation server[1]. For the leaderboard, CIDEr-c40, specially designed for image captioning, is the default ranking metric, which is more convincing than CIDEr-c5, as shown in Vedantam et al. [50] that CIDEr achieves higher correlation with human judgment when more reference sentences are given. The results of our approach and the top-performing published works on the leaderboard are reported on Table 2. As we can see, we outperform all the works in terms of CIDEr-c40 and rank the 1st.

### 3.3 Results of Grounding Performance

We evaluate the grounding performance using both automatic metrics and human evaluations.

**Automatic Evaluation**    As shown in Table 1, by incorporating our method into the baselines, both $F1_{all}$ and $F1_{loc}$ increase up to 30% and 31%, respectively. This demonstrate that our approach can not only help to generate the correct word but also attend to the proper image region at the same time.

**Human Evaluation**    Since the grounding performance reflects the interpretability of the model, it is necessary to conduct human based evaluations. Therefore we introduce human evaluation to compare the Dynamic Prophet Attention (DPA) with baselines. We randomly select 500 samples from the Flickr30k Entities and MSCOCO test sets, that is 250 samples from each dataset. We recruit 10 workers to compare the perceptual quality of the grounding between our approach and baselines independently. The results in Table 3 show that our approach wins in more samples than all baselines.

Overall, our approach outperform baselines under all metrics in both captioning and grounding.

Table 3: Grounding performance of human evaluation on the Flickr30k Entities and the MSCOCO image captioning datasets for comparing our approach with baselines.

| Datasets | vs. Models | Baseline wins (%) | Tie (%) | w/ DPA wins (%) |
|---|---|---|---|---|
| Flickr30k Entities | Up-Down | 19.6 | 46.8 | **33.6** |
| | GVD | 23.6 | 44.4 | **32.0** |
| MSCOCO | Up-Down | 22.0 | 40.4 | **37.6** |
| | AoANet | 26.4 | 38.8 | **34.8** |

Table 4: Quantitative analysis of our approach. We conduct the analysis on the Up-Down [2]. * denotes our own implementation.

| Methods | $\lambda$ | | | Flickr30k Entities | | | | MSCOCO | | | |
|---|---|---|---|---|---|---|---|---|---|---|---|
| | | $F1_{loc}$ | $F1_{loc}$ | BLEU-4 | METEOR | CIDEr | SPICE | BLEU-4 | METEOR | ROUGE-L | CIDEr | SPICE |
| Up-Down [2] | - | 4.53 | 13.0 | 27.3 | 21.7 | 56.6 | 16.0 | 36.3 | 27.7 | 56.9 | 120.1 | 21.4 |
| Baseline* | - | 4.19 | 12.1 | 26.4 | 21.5 | 57.0 | 15.6 | 36.7 | 27.9 | 57.1 | 123.5 | 21.3 |
| w/ CPA | 0.05 | 4.96 | 13.8 | 26.7 | 21.6 | 58.8 | 15.9 | 37.4 | 28.3 | 57.6 | 126.3 | 21.6 |
| w/ DPA | 0.05 | 5.21 | 14.9 | 27.1 | 22.1 | 60.3 | 16.0 | 38.2 | 28.8 | 56.9 | 128.1 | 21.9 |
| w/ DPA | 0.1 | **5.45** | **15.3** | **27.2** | **22.3** | **60.8** | **16.3** | **38.6** | **29.1** | **58.3** | **129.0** | **22.2** |
| w/ DPA | 0.2 | 4.47 | 13.5 | 26.5 | 21.8 | 58.4 | 15.8 | 37.8 | 28.5 | 57.9 | 125.8 | 21.7 |
| w/ DPA | 0.3 | 2.82 | 8.2 | 26.0 | 21.5 | 57.1 | 15.4 | 36.6 | 27.7 | 57.2 | 121.3 | 21.0 |
| w/ DPA | 1 | 0.26 | 1.04 | 23.8 | 20.1 | 50.6 | 14.1 | 35.7 | 27.0 | 55.7 | 114.8 | 19.7 |
| w/o attention | - | - | - | 25.4 | 20.5 | 54.0 | 14.8 | 35.9 | 27.3 | 56.4 | 115.7 | 20.3 |

## 4   Analysis

In this section, we conduct analysis from different perspectives to better understand the proposed Prophet Attention.

**Quantitative Analysis**    In this section, we conduct the quantitative analysis on the representative model, i.e., Up-Down [2], to evaluate the contribution of each component in our approach.

*Comparison between CPA and DPA*    Table 4 shows that both variants of our Prophet Attention (CPA and DPA) can promote the baseline over all metrics substantially, which proves our arguments. However, compared with DPA, CPA's performance is relatively lower. This may due to that CPA introduced confusing visual information for the attribute words and noisy visual information for non-visual words. To verify this hypothesis, we first conduct the human evaluation to compare the "w/ DPA" and "w/ CPA" in terms of the object words, e.g., *car*, attribute words, e.g., *wooden*, and relation words, e.g., *sit*. The results on 500 samples from MSCOCO dataset show that "w/ DPA" performs better than "w/ CPA" over the three categories, especially in terms of attribute words (see Table 5). Besides, our experimental results also show that if we does not MASK our attention model when the output words are non-visual words, it will cause a performance decrease, i.e., a 0.5 drop in CIDEr and a 0.3 drop in SPICE. These experimental results demonstrate the effectiveness of our proposed Dynamic Prophet Attention (DPA).

*Effect of $\lambda$*    Table 4 shows that when $\lambda$ is larger than 0.1, both the captioning and grounding performance will decrease as $\lambda$ increases. We take the CPA as an example to explain the phenomenon. For each timestep $t$, the attentional weights are regularized to approximate the attention weights in next timestep $t + 1$, which are further regularized by the attention weights in the timestep $t + 2$. Through such nested regularization, when $\lambda$ is set to large values, the attention weights tends to bias towards the attention weights of the last token in this sequence. To verify this, we set a large value of $\lambda = 1$ and observe that it has the same captioning performance as "w/o attention" model, with an extremely low grounding accuracy (see Table 4).

*Analysis on the Loss Function*    We further apply the L2 norm and KL divergence to the Up-Down model. The results with L1 norm, L2 norm and KL divergence are 129.0, 128.2 and 126.9 CIDEr which all outperform the baseline model (123.5 CIDEr). It also shows that L1 norm achieves the best performance and all loss functions are viable in practice with improved performance, which proves the effectiveness and robustness of our approach.

Table 5: Results of human evaluation on the MSCOCO dataset in terms of object, relationship and attribute categories.

| Categories | "w/ CPA" wins (%) | Tie (%) | "w/ DPA" wins (%) |
|---|---|---|---|
| Object | 25.8 | 44.6 | **29.6** |
| Relationship | 25.0 | 46.6 | **28.4** |
| Attribute | 21.2 | 43.0 | **35.8** |

Table 6: Results of paraphrase and video captioning tasks.

| Methods | Paraphrase | | Video Captioning |
|---|---|---|---|
| | BLEU | METEOR | CIDEr |
| Baseline | 29.2 | 23.5 | 48.9 |
| w/ DPA | **36.5 (+7.3)** | **26.8 (+3.3)** | **52.2 (+3.3)** |

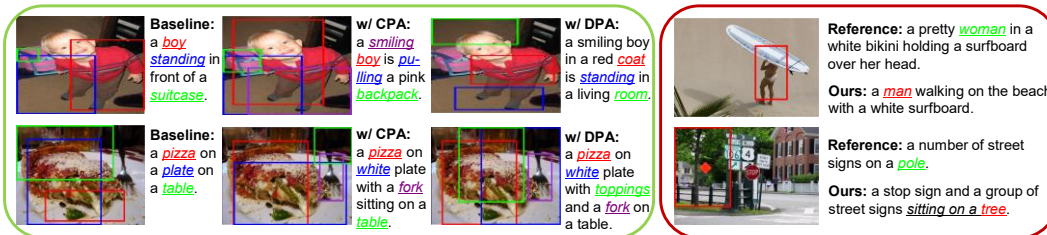

Figure 3: Examples of the generated captions and corresponding visual grounding regions. The left plot and right plot show the correct examples and the error analysis of our approach, respectively. Please view in color. For each marked generated word, we show the top-1 attended image region. As we can see, our approach could generate more desirable captions and correctly ground the image region with generated word. For the error example, although our approach generates an unfavorable caption, it could still select the correct image region.

**Generalization Analysis**     In addition to image captioning, the Prophet Attention can also be applied to other similar generation tasks. Therefore, we further conduct experiments on the MSCOCO dataset for paraphrase generation task [15, 31, 43] and the MSR-VTT dataset for video captioning task [6, 52]. For detailed descriptions of these two tasks and the implementation details, please refer to our supplementary materials.

*Paraphrase*     Paraphrases convey the same meaning as the original sentences or text, but with different expressions in the same language. Paraphrase generation aims to synthesize paraphrases of a given sentence automatically [31, 36, 40]. For the experiments, we implement the standard sequence-to-sequence with attention model (LSTM-Attention) [3] and use the default setting provided by *OpenNMT* [23] on MSCOCO dataset as our baseline. In Table 6, as we can see, by using our approach, we can achieve an improved performance of 7.3 BLEU score and 3.3 METEOR score.

*Video Captioning*     Compared with image captioning, the target of video captioning is the video clip, i.e., an ordered sequence of images, thus it is relatively more challenging, as there are more visual features needed to be considered, such as motion features, audio features and the temporal dynamics information. For the experiments, we implement the Up-Down [2] on video captioning task as the baseline. Table 6 reports the CIDEr score [50], which is specifically designed to evaluate captioning systems. As we can see, our approach can improve the performance of baseline substantially, outperforming the current state-of-the-art model MGSA (50.1 CIDEr) [6], which further demonstrates the effectiveness of our proposed approach.

**Qualitative Analysis**     In the left sub-figure of Figure 3, we list some correct examples on MSCOCO image captioning dataset to better understand our approach intuitively. As we can see, because our CPA and DPA could employ the future information to guide the attention model to select the correct regions for generating the corresponding words, we help the baseline model to ground more accurate image regions and generate more desirable captions. Especially, the DPA could bring better grounding accuracy in terms of attribute words, such as *white* than that of CPA.

We also list some bad cases in the right sub-figure of Figure 3 to provide insights on how our approach may be improved. There are two representative types of errors, i.e., visual similarity and location relationship. As we can see, in the first example, our model misidentify the *woman* as a *man* due to their visual similarity. However, humans can find that the person in the image is wearing a *bra*. In the second example, our approach describes an irrelevant location relationship for the objects, i.e., *a stop sign [. . . ] sitting on a tree*. Although our approach generates unfavorable captions, it still could correctly ground the image region with the generated word.

# 5   Related Work

**Image Captioning**    In recent years, a large number of neural systems have been proposed for image captioning [2, 10, 17, 20, 27, 38, 51, 53, 54]. The state-of-the-art approaches [2, 10, 17, 20, 38] depend on the encoder-decoder framework to translate the image to a descriptive sentence. Specifically, the encoder network computes visual representations for the image and the decoder network generates a target sentence based on the visual representations. To allow a more efficient use of the visual representations, a series of attention models have been proposed and achieve great success in multiple sequence-to-sequence learning tasks [3, 34, 37, 49, 53]. For image captioning, Xu et al. [53] proposes the visual attention to help the model to focus on the most relevant image regions instead of the whole image; in Lu et al. [32], an adaptive attention model is designed to decide when to employ the visual attention. There are other attention mechanisms such as: spatial and channel-wise attention [5], semantic attention [56] and attention on attention [20]. However, these methods calculate attentional weights without explicitly knowing the word it should ground.

**Attention Supervision**    Recently, some works [25, 57, 59] found that adding supervision to attention model is beneficial for captioning model. More specifically, various approaches [25, 33, 48, 59] have been proposed for various vision-and-language problems, e.g., referring expression [30, 30] and grounding visual explanations [16]. However, for image captioning, these existing methods [25, 33, 59] explicitly provide a series of ground truth bounding box annotations for training, while our model does not require such additional labeling. Instead, our model learns the grounded attentional weights implicitly from the descriptions of images in a manner similar to self-supervision without bounding box annotations. Therefore our approach can be directly applied to existing image captioning models.

**Incorporating Future Information**    Most recently, Chen et al. [8], Duan et al. [13], Qin et al. [44], Ren et al. [46] and Ma et al. [35] attempted to exploit the future information to boost the performance for sequence-to-sequence learning systems. However, their modelings are different from ours. Specifically, to exploit the future information, at each time step, Chen et al. [8] first adopt the fine-tuned BERT [12] to encode the words that will be generated in the future to acquire the future information. And then the obtained future information is exploited as an extra supervision to guide the current word generation based on the previous generated words in conventional sequence-to-sequence models. In this way, Chen et al. [8] can incorporate the future information for predicting the present to improve the performance of text generation. For Duan et al. [13], Qin et al. [44] and Ren et al. [46], given a time stamp in sequence generation, in addition to current target, they further predict the future words, i.e., Duan et al. [13], Qin et al. [44] further predict target word one more step ahead, and Ren et al. [46] predicts the rest of the sequence. The prediction accuracy of future targets and current target are combined together in parameter optimization. While in our proposed approach, we use the hidden states of future time stamps to explicitly guide the attention calculation of current one. In other words, we aim to better ground current target word to proper image regions which alleviates the attention deviation problem, resulting in improved performance of both grounding and captioning. One of them [35] also conduct a similar effort. The cyclical training regimen proposed in Ma et al. [35] is similar to the idea of our Constant Prophet Attention. In particular, Ma et al. [35] only consider the current target to model the future information, while we leverage the bidirectional knowledge to enhance the modeling of the future information. Besides, we further propose the Dynamic Prophet Attention and the Prophet Knowledge Distillation trick for better exploiting the future information.

# 6   Conclusions

In this work, we focus on correctly grounding the image regions with generated words in the attention model, without any grounding annotations. To this end, we propose the Prophet Attention, which is similar to the form of self-supervision for calculating attentional weights based on future information, and force the attention model to learn to correctly ground each output word to proper image regions. Extensive experiments and analysis demonstrate the effectiveness and the generalization capabilities of our approach to a wide range of models and language generation tasks. Specifically, our proposed approach consistently boosts the captioning performance of the baselines under all metrics across the board and significantly improves the grounding accuracy without the ground-truth of grounding annotation. More encouragingly, we set new state-of-the-art performance on the Flickr30k Entities and the MSCOCO datasets for image captioning.

# 7 Broader Impact

Our work aims to improve both the captioning and grounding performance of image captioning systems, promoting the real-word application of image captioning, such as visual retrieval, human-robot interaction and visually impaired people assistance. Furthermore, we can also improve the model interpretability and transparency. However, the training of our framework relies on large volume of image-caption pairs, which are not easily obtained in the real-world. Therefore, it requires specific and appropriate treatment by experienced practitioners.

## Acknowledgments

This work is partly supported by National Key R&D Program of China (2020AAA0105200), Beijing Academy of Artificial Intelligence (BAAI), National Engineering Laboratory for Video Technology - Shenzhen Division, and Shenzhen Municipal Development and Reform Commission (Disciplinary Development Program for Data Science and Intelligent Computing). Special acknowledgements are given to AOTO-PKUSZ Joint Lab for its support. We thank all the anonymous reviewers for their constructive comments and suggestions. Xu Sun and Yuexian Zou are the corresponding authors of this paper.

## Footnotes

[1] https://competitions.codalab.org/competitions/3221#results

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
