[Supplementary Material]

# Appendix of Prophet Attention

**Fenglin Liu[1], Xuancheng Ren[2], Xian Wu[3], Shen Ge[3], Wei Fan[3], Yuexian Zou[1,4], Xu Sun[2,5]**

[1]ADSPLAB, School of ECE, Peking University, Shenzhen, China
[2]MOE Key Laboratory of Computational Linguistics, School of EECS, Peking University
[3]Tencent, Beijing, China   [4]Peng Cheng Laboratory, Shenzhen, China
[5]Center for Data Science, Peking University
{fenglinliu98, renxc, zouyx, xusun}@pku.edu.cn
{kevinxwu, shenge, Davidwfan}@tencent.com

| # | User | Entries | Date of Last Entry | BLEU-1 | | BLEU-2 | | BLEU-3 | | BLEU-4 | | METEOR | | ROUGE-L | | CIDEr-D | |
|---|------|---------|--------------------|--------|---|--------|---|--------|---|--------|---|--------|---|--------|---|--------|---|
| | | | | c5 ▲ | c40 ▲ | c5 ▲ | c40 ▲ | c5 ▲ | c40 ▲ | c5 ▲ | c40 ▲ | c5 ▲ | c40 ▲ | c5 ▲ | c40 ▲ | c5 ▲ | c40 ▲ |
| 1 | Prophet | 1 | 06/02/20 | 0.818 (5) | 0.963 (1) | 0.665 (6) | 0.912 (1) | 0.519 (6) | 0.832 (1) | 0.398 (8) | 0.733 (1) | 0.296 (3) | 0.393 (2) | 0.594 (5) | 0.751 (2) | 1.304 (3) | 1.337 (1) |
| 2 | Tsinghua-Samsung | 1 | 05/27/20 | 0.821 (1) | 0.960 (3) | 0.669 (2) | 0.908 (3) | 0.525 (2) | 0.827 (4) | 0.403 (3) | 0.727 (5) | 0.299 (1) | 0.395 (1) | 0.597 (2) | 0.756 (1) | 1.319 (1) | 1.336 (2) |
| 3 | Yingwei.Pan | 5 | 03/23/20 | 0.819 (3) | 0.957 (8) | 0.669 (3) | 0.905 (7) | 0.524 (4) | 0.825 (7) | 0.403 (4) | 0.724 (8) | 0.296 (2) | 0.392 (3) | 0.595 (4) | 0.750 (4) | 1.311 (2) | 1.335 (3) |
| 4 | luo3300612 | 4 | 05/21/20 | 0.816 (8) | 0.961 (2) | 0.665 (7) | 0.909 (2) | 0.519 (7) | 0.828 (2) | 0.397 (10) | 0.729 (2) | 0.294 (7) | 0.388 (9) | 0.591 (12) | 0.744 (16) | 1.303 (4) | 1.325 (4) |
| 5 | Meshed-Memory-Transformer | 1 | 11/13/19 | 0.816 (7) | 0.960 (4) | 0.664 (8) | 0.908 (4) | 0.518 (11) | 0.827 (3) | 0.397 (11) | 0.728 (3) | 0.294 (8) | 0.390 (5) | 0.592 (8) | 0.748 (7) | 1.293 (7) | 1.321 (5) |
| 6 | DiMBERT | 1 | 05/24/20 | 0.816 (10) | 0.959 (5) | 0.664 (9) | 0.907 (5) | 0.518 (9) | 0.826 (5) | 0.398 (9) | 0.728 (4) | 0.293 (9) | 0.389 (6) | 0.592 (7) | 0.748 (8) | 1.289 (10) | 1.317 (6) |
| 7 | KingSoft_AILAB | 1 | 10/24/19 | 0.819 (4) | 0.957 (7) | 0.670 (1) | 0.906 (6) | 0.527 (1) | 0.825 (6) | 0.406 (1) | 0.725 (6) | 0.294 (5) | 0.388 (10) | 0.598 (1) | 0.750 (5) | 1.298 (6) | 1.317 (7) |
| 8 | Bridging_the_Gap | 1 | 05/25/20 | 0.809 (26) | 0.957 (6) | 0.657 (21) | 0.904 (9) | 0.512 (21) | 0.822 (10) | 0.393 (21) | 0.722 (9) | 0.295 (4) | 0.390 (4) | 0.592 (10) | 0.746 (9) | 1.290 (8) | 1.316 (8) |
| 9 | IVA-HUAWEI | 1 | 07/29/19 | 0.816 (9) | 0.956 (12) | 0.666 (5) | 0.903 (10) | 0.521 (5) | 0.822 (9) | 0.401 (5) | 0.722 (10) | 0.293 (11) | 0.389 (8) | 0.594 (6) | 0.749 (6) | 1.290 (9) | 1.314 (9) |
| 10 | MIL-HDU | 3 | 08/13/19 | 0.817 (6) | 0.956 (10) | 0.668 (4) | 0.905 (8) | 0.524 (3) | 0.824 (8) | 0.404 (2) | 0.724 (7) | 0.294 (6) | 0.389 (7) | 0.596 (3) | 0.750 (3) | 1.300 (5) | 1.309 (10) |

Figure 1: Leaderboard captioning performance on the online MSCOCO evaluation server. c5 and c40 mean comparing to 5 references and 40 references, respectively. As we can see, at the time of submission (2 June 2020), we outperform all the top-performing works in terms of CIDEr-c40, which is the default ranking score, and rank the 1st.

## A   Leaderboard

Following common practice [1, 12, 15, 28], we also submit our approach model to online MSCOCO evaluation server[1]. For the leaderboard, CIDEr-c40, specially designed for image captioning, is the default ranking metric, which is more convincing than CIDEr-c5, as shown in Vedantam et al. [25] that CIDEr achieves higher correlation with human judgment when more reference sentences are given. The results of our approach and all the top-performing works, including unpublished works, on the leaderboard are shown in Figure 1. As we can see, we outperform all the works in terms of CIDEr-c40, which is the default ranking score in the leaderboard, and rank the 1st. It demonstrates the effectiveness of our proposed approach.

# B  Task and Implementation Details

## B.1  Paraphrase

Paraphrases convey the same meaning as the original sentences or text, but with different expressions in the same language. Paraphrase generation aims to synthesize paraphrases of a given sentence automatically. This is a fundamental natural language processing task, and it is important for many downstream applications [17, 19, 21].

We train the models on the MSCOCO dataset[2] [6], which is a large-scale captioning dataset which contains human annotated captions of over 120K images. This dataset was used previously to evaluate paraphrase generation methods [9, 23]. In the MSCOCO dataset, each image has five captions from five different annotators. Annotators describe the most obvious object or action in an image, which makes this dataset very suitable for the paraphrase generation task. This dataset comes with separate subsets for training and validation: Training set contains over 82K images and validation set contains over 40K images. From the five captions accompanying each image, we randomly omit one caption and use the other four as training instances to create paraphrase pairs. In order to compare our results with previous work [9, 23], 20K instances are randomly selected from the data for testing, 10K instances for validation and remaining data over 320K instances for training.

We use NLTK [4] to tokenize the sentences and keep words that appear more than 10 times in our vocabulary. Following Gupta et al. [9] and Prakash et al. [23], we reduce those captions to the size of 15 words (by removing the words beyond the first 15) for the MSCOCO dataset. For the experiments, we implement the standard sequence to sequence with attention model (LSTM-Attention) [2] and use the default setting provided by *OpenNMT* [11]. BLEU [20] and METEOR [3] are used to evaluate the performance of models.

## B.2  Video Captioning

Compared with image captioning, the target of video captioning is the video clip, i.e., an ordered sequence of images, thus it is relatively more challenging, as there are more visual features needed to be considered, such as motion features, audio features and the temporal dynamics information. Our reported results are evaluated on the popular Microsoft Research Video to Text (MSR-VTT)[3] [27] dataset. The dataset contain 10,000 video clips, and each video is paired with 20 annotated sentences. We use the official splits to report our results. Thus, there are 6513, 497 and 2,990 video clips in training set, validation set and test set, respectively. Following common practice, we replace caption words that occur less than 3 times in the training set with the generic unknown word token [UNK], plus with a [MASK] token, resulting in a vocabulary of 10546 words for MSR-VTT. The metric CIDEr, which is built upon on n-gram matching, is used in our tests for performance evaluation. It is widely used and could be calculated by the MSCOCO captioning evaluation toolkit [6].

# C  Experimental Settings

For our proposal, $\lambda$ is set to 0.1, according to the average performance on the validation set. For fair comparisons, since our approach aims to force the conventional attention model to learn to ground each output word to source information and is an augmentation to the existing models, we keep the inner structure of the baselines untouched and preserve the original settings. Our code is implemented in PyTorch [22]. All re-implementations and our experiments were ran on V100 GPUs.

In conventional image captioning works [5, 8, 14, 16, 18, 26, 29], they first pre-train the models by minimizing the cross-entropy loss. Specifically, given a target ground truth sequence $y_{1:T}^*$ and a captioning model with parameters $\theta$, the goal is to minimize the cross entropy loss:

$$\mathcal{L}_{\text{CE}}(\theta) = -\sum_{t=1}^{T} \log\left(p_\theta\left(y_t^*|y_{1:t-1}^*\right)\right).\tag{1}$$

Then, for image captioning task, common practice [1, 13, 15, 28, 30] further adopts CIDEr-based training objective using reinforcement training [24] to improve the performance of image captioning

models. In this case, the cross-entropy loss is used to pre-train the model, after which the goal is to minimize the negative expected score as:

$$\mathcal{L}_{\text{RL}}(\theta) = -\mathbb{E}_{y_{1:T} \sim p_\theta}[r(y_{1:T})] \tag{2}$$

where $r$ is the score function (i.e., CIDEr); $y_{1:T}$ denotes the caption generated by the current model. Following the Self-Critical Sequence Training [24] (SCST), the gradient of $\mathcal{L}_{\text{RL}}(\theta)$ can be approximated by

$$\nabla_\theta \mathcal{L}_{\text{RL}}(\theta) \approx -(r(y_{1:T}^s) - r(\hat{y}_{1:T}))\nabla_\theta \log p_\theta(y_{1:T}^s) \tag{3}$$

where $r(y_{1:T}^s)$ is the score of a sampled caption $y_{1:T}^s$ and $r(\hat{y}_{1:T})$ suggests the baseline score of a caption which is generated by the current model using greedy decode. Through this gradient, sampled captions with higher CIDEr scores are more likely to be generated by the model because their corresponding probabilities are increased.

## D  Prophet Knowledge Distillation for Image Captioning

In this section, we introduce a simple trick, i.e., Prophet Knowledge Distillation (PKD), which is inspired by the work of Chen et al. [7], to better exploit the future information $y_{1:T}$ during the reinforcement training to boost the performance of image captioning models.

In implementation, at each time step $t$, we use the $y_{1:T\setminus t}$ to predict the current target word $y_t$:

$$\beta_t = f_{\text{Att}}(h_t, Y) = \text{softmax}\left(w_\beta \tanh\left(W_h^\beta h_t \oplus W_Y^\beta Y\right)\right), \quad c_t^\beta = Y\beta_t^{\text{T}},$$

$$y_t \sim p_t = \text{softmax}\left(W_p^\beta [h_t; c_t^\beta] + b_p^\beta\right), \tag{4}$$

where $Y = W_e y_{1:T\setminus t}$ ($W_e$ is the word embedding parameters); the $w_\beta$, $W_h^\beta$, $W_Y^\beta$, $W_p^\beta$ and $b_p^\beta$ are the new learnable parameters; $h_t$ is the current hidden state of the conventional image captioning model.

Now, we can adopt the following cross-entropy loss to further optimize the full image captioning model with the parameters $\theta$, including the new parameters introduced by the PKD:

$$\mathcal{L}'_{\text{CE}}(\theta) = -\sum_{t=1}^{T} \log\left(p_\theta\left(y_t | y_{1:T\setminus t}\right)\right). \tag{5}$$

During the reinforcement training, we combine the loss $\mathcal{L}_{\text{RL}}(\theta)$ in Eq. (2) and the loss $\mathcal{L}'_{\text{CE}}(\theta)$ in Eq. (5) to train the full image captioning model with the parameters $\theta$:

$$\mathcal{L}_{\text{Full}}^{\text{RL}}(\theta) = \mathcal{L}_{\text{RL}}(\theta) + \mathcal{L}'_{\text{CE}}(\theta) = -\mathbb{E}_{y_{1:T} \sim p_\theta}[r(y_{1:T})] - \sum_{t=1}^{T} \log\left(p_\theta\left(y_t | y_{1:T\setminus t}\right)\right), \tag{6}$$

In this way, we can better exploit the future information to boost the performance of image captioning models. The improvements in Table 1 prove our arguments and demonstrate the effectiveness of the introduced PKD.

Table 1: The analysis of the Prophet Knowledge Distillation (PKD). We perform the analysis on the MSCOCO image captioning dataset. § and † denote our own implementation without the PKD trick and statistically significant results (t-test with $p < 0.01$), respectively.

| Methods | MSCOCO | | | | |
| --- | --- | --- | --- | --- | --- |
| | BLEU-4 | METEOR | ROUGE-L | CIDEr | SPICE |
| Up-Down [1] | 36.3 | 27.7 | 56.9 | 120.1 | 21.4 |
| Up-Down§ | 36.5 | 28.0 | 57.0 | 120.9 | 21.5 |
| w/ PKD | **36.7**† | **27.9**† | **57.1**† | **123.5**† | **21.3**† |
| AoANet [10] | 38.9 | 29.2 | 58.8 | 129.8 | 22.4 |
| AoANet§ | 38.3 | 28.6 | 58.6 | 128.2 | 22.1 |
| w/ PKD | **38.8**† | **29.0**† | **58.7**† | **129.6**† | **22.6**† |

## Footnotes

[1]https://competitions.codalab.org/competitions/3221#results

[2] http://mscoco.org/

[3] http://ms-multimedia-challenge.com/2017/dataset