[Reviews · NeurIPS 2020]

Review 1

Summary and Contributions: From the motivation that existing methods on image captioning methods have a "deviated focus" problem, which attention-based decoder calculate the attention weights used on previous words instead of the one to be generated, this submission proposes a method to regularize such attention weight using future attention weights

Strengths: - Easy to follow writing. - Thorough experiments with ablation study - SOTA performance

Weaknesses: - While I enjoyed the introduction and motivation, the implementation of solving "deviated focus" problem seems straight-forward (i.e., average of future attention weights, POS tag words and split the cases) - It would be nice to have ablation on loss function for attention regularization using L2 norm, KL divergence, etc instead of L1 norm. Why did you choose L1 norm? ============================================================ Author feedback clarifies my questions.

Correctness: It seems correct

Clarity: Yes

Relation to Prior Work: Yes

Reproducibility: No

Additional Feedback:


Review 2

Summary and Contributions: This paper focuses on the "deviated focus" problem on attention-based image captioning models. Specifically, the attention weights are calculated based on previous words instead of the one to be generated. Thus, the authors calculate attention weights from the "future words" to regularize the "deviated" attention. Extensive results on Flickr30k Entities and MSCOCO datasets have demonstrated the effectiveness of the Prophet Attention.

Strengths: 1. The whole paper is well written and easy to follow. 2. The proposed Prophet Attention mechanism is model-agnostic and can be easily incorporated into different image captioning models and other tasks (ie, paraphrase and video captioning). 3. The model achieves a new state-of-the-art performance on two benchmarks.

Weaknesses: ========= Post-rebuttal Edit ============= After reading the authors' response and all other reviewers' comments, I choose to keep my rating as 6 (Weak Accept) for the following reasonings: 1) The proposed paper is well-written and all strong results have demonstrated the effectiveness. 2) The whole model is simple, straightforward, and experiments-oriented. More comprehensive considerations about different situations or words (like the mentioned verb) can be better. ==================================== 1. The ideas are partially not convincing, the attention for some words should not only from the future, but also from the history. As mentioned in Eq. (9), the Dynamic Propohet Attention considers three types of words: noun phrase, non-visual word, and others. For example in Figure 1, when generate some verb "holding", based on the current formulation, they belong to the "others" category. And the \hat{\alpha} (cf. Eq(5)) come from the the next future word. However, intuitively, it is more reasonable to focus on both the subject and object of "holding", \ie, "a woman" and "a yellow umbrella". Analogously, for "wearing", the targeted attention should focus on "a woman" and "a yellow coat".

Correctness: Yes. Although I think the attention of some words should focus on both history and future (see Weaknesses part), in many cases, the attention should be consistent with the current generated words.

Clarity: Yes. This paper is well written and all figures are well-illustrated.

Relation to Prior Work: Yes. This paper have a clearly discussion about the difference between existing attention-based models and attention supervision on other vision-and-language tasks.

Reproducibility: Yes

Additional Feedback: 1. For the first visualization results in Figure 3, the model with DPA can predict smiling with the purple box, is it should be attend to the whole boy (ie, "a smiling boy")?


Review 3

Summary and Contributions: Different from previous attention-based models usually use the hidden state of current input to attend to the image regions, this paper proposes a Prophet Attention to enable attention models to correctly ground words to be generated to proper image regions. The proposed model achieves great success on image captioning. -----------update after rebuttal -------------------- Authors have addressed all my concerns in the rebuttal and I vote for acceptance of this submission.

Strengths: The motivation for this paper is clear and soundness. The authors identified a significant problem of current attention-based models (for image captioning) and propose a simple 'trick' to address it. The proposed method is easy to follow and implement. And several experiments have been done to prove its effectiveness. The paper is clear and easy to follow.

Weaknesses: They are not weaknesses but I have some concerns: 1. I do agree with the authors that the 'attention' should not be calculated based on the previous states, instead, it should use the 'future' states. However, according to the setting of hyper-parameter 'lambda', the model only works well when it equals to 0.1, which is a rather small value. This may suggest that this 'modification' of the attention mechanism is not significant. I can accept the explanation that a large 'lambda' will lead to the attention weight bias towards the last token in the sequence, but whether this suggests the 'nested' regularization mechanism proposed in this paper is not appropriate or even a wrong direction? 2. It is OK to say the proposed model outperforms all the published state-of-the-art models, however, I don't think it is appropriate to say it achieves the 1st place on the leaderboard. It only achieves the best results when C40 references are used. And it is not clear whether other models use ensemble or not. 3. This will be an excellent work if authors can show it also works on other attention-based models, on non-caption generation tasks.

Correctness: The method is simple but looks effective.

Clarity: yes, very clear and easy to follow.

Relation to Prior Work: yes, the differences are clear.

Reproducibility: Yes

Additional Feedback:


Review 4

Summary and Contributions: In this paper, the author focus on correctly grounding the image regions with generated words in the attention model, without any grounding annotations, additional parameters and extra inference computations. Importantly, they propose the Prophet Attention, which is similar to the form of self-supervision for calculating attentional weights based on future information, and force the attention model to learn to correctly ground each output word to proper image regions. In my view, the contributions are: 1. a new attention mechanism to correctly ground words to proper image regions. 2. achieves a new good performance. 3. this strategy is general and can be applied to other similar areas. In particular, I am glad that this work achieves a new 1st place on the COCO leaderboard.

Strengths: 1. As written above, this method achieves a good performance. 2. Figure 2 is pretty clear for understanding. 3. It is robust to perform experiment on baselines: Up-Down, GVD and AoANet. 4. Experiment evaluation are adequate and compact. 5. This method can be applied to other similar areas.

Weaknesses: 1. The paper is experimental oriented, and there is not enough theoretical analysis and grounding. 2. The paper does not explain the influence of introducing structure on time and lacks relevant analysis. How much the model slow down? 3. The idea is not novel, since incorporating future information has been studied and this paper has not presented. 4. There are many grammatical and clerical errors in the text.

Correctness: Pretty correct.

Clarity: not up to much, there are a lot of mistakes, e.g. line 220 Table 4 should be Table 6.

Relation to Prior Work: Incorporating future information is not new in image captioning task, for example, [1] [2][3], however, the author has not compared. I think author should illustrate the difference. [1] Modeling Future Cost for Neural Machine Translation [2] Qin, Yu, et al. "Look back and predict forward in image captioning." Proceedings of the IEEE Conference on Computer Vision and Pattern Recognition. 2019. [3] Ren, Zhou, et al. "Deep reinforcement learning-based image captioning with embedding reward." Proceedings of the IEEE conference on computer vision and pattern recognition. 2017.

Reproducibility: Yes

Additional Feedback: This is a nice article and will make a good contribution to the image captioning community. I hope the author can select several weakness as feedback.

[Author Response · NeurIPS 2020]

We thank all the reviewers for the helpful comments. We will revise the paper to address your concerns.

**R1-Q1: The implementation seems straight-forward and the ablation analysis on the loss function.**

In the beginning, based on the Up-Down model, we have attempted to implement the Constant Prophet Attention
with more complex models, i.e., single-head dot-product attention (120.2 CIDEr), multi-head dot-product attention
(118.8 CIDEr), bilinear attention (122.8 CIDEr) and attention-based LSTM (121.7 CIDEr). However, they all lower the
performance of Up-Down base model (123.5 CIDEr on MSCOCO) and introduce extra model parameters and inference
computations. While our current implementation can boost the performance without introducing any additional
parameters or slowing down inference computations. For ablation analysis, our preliminary experiments showed that
using L1 norm improves the performance. Thus we kept using L1 norm in the rest of experiments. We follow your
constructive advice to apply the L2 norm and KL divergence to the Up-Down model. The results with L1 norm, L2
norm and KL divergence are 129.0, 128.2 and 126.9 CIDEr which all outperform the original Up-Down model. It shows
that L1 norm achieves the best performance and all loss functions are viable in practice with substantially improved
performance. We will conduct a systematic comparison between various loss functions in the next revision.

**R2-Q1: The attention for some words should not only from the future, but also from the history.**

For noun phrases, our approach uses both history ($i < t$) and future ($j > t$) information (see Eq. (5) and Eq. (9)).
For other cases like verbs, we did not consider the explicit history information. However in the process of sequence
generation, the hidden state could contain implicit information in previous time stamps. Nevertheless, we will attempt to
adopt the scene graph, which is able to extract triples from a sentence or an image. The triples like *<subject-verb-object>*,
e.g., *<woman-holding-umbrella>*, can provide more explicit history information.

**R2-Q2: The first visualization results of DPA in Figure 3 of our paper.**

In this example, the top-1 attended region for the noun phrase "a smiling boy" is '*mouth*'. Therefore, in the context of
DPA, the attended region of the word "smiling" is the same as the noun phrase "a smiling boy".

**R3-Q1: The value of the hyper-parameter $\lambda$.**

To choose a proper value of $\lambda$, we follow the setting of the regularization losses like weight decay and adversarial loss
which choose small $\lambda$ values. For example, the weight of L2 regularization is "tiny", often set as low as 1e-4, but has a
positive effect on the model training. We also evaluate the performance of different settings of $\lambda$ in Table 4 to ensure
that the degree of our regularization is tuned to match the model training.

**R3-Q2: The evaluation on non-caption generation tasks and the 1st place on the leaderboard.**

Thanks for your great suggestion for the evaluation on non-caption generation tasks. It is a good future direction and we
will work on it. For the leaderboard, CIDEr-c40, specially designed for captioning, is the default ranking metric, which
is more convincing than CIDEr-c5, as shown in Vedantam et al. [2015] that CIDEr achieves higher correlation with
human judgment when more reference sentences are given. It is worth noticing that most current published top rankers
use ensemble. However, it's true that we are not aware of whether other submissions use ensemble or not, if they did
not publish their approaches. We will tone down our voice in the next revision.

**R4-Q1: How much the model slows down?**

In the inference stage, our approach does not incur extra computational cost. In the training stage, extra computations
are mainly introduced by the calculation of L1 loss (see Eq. (6)) and the Prophet Attention (see Eq. (9)). We find that
our approach slows down the training procedure around 4%. We will provide detailed information in the next revision.

**R4-Q2: The grammatical and clerical errors. What is the difference between our work and previous works?**

Thank you very much! We will polish this paper and fix the grammatical and clerical errors. For these three related
works, we will cite and discuss them in detail. Although [1,2,3] attempted to exploit the future information, their
modelings are different from ours. Given a time stamp in sequence generation, in addition to current target, [1,2,3]
introduce new model parameters to further predict the future words. Specifically, [1,2] further predict target word one
more step ahead, and [3] predicts the rest of the caption. The prediction accuracy of future targets and current target are
combined together in parameter optimization. While in our proposed approach, we use the hidden states of future time
stamps to explicitly guide the attention calculation of current one without introducing additional model parameters or
inference computations. In other words, we aim to better ground current target word to proper image regions which
alleviates the attention deviation problem, resulting in improved performance of both grounding and captioning.

# References

R. Vedantam, C. L. Zitnick, and D. Parikh. Cider: Consensus-based image description evaluation. In *CVPR*, 2015.


[Meta-Review · NeurIPS 2020]

All the reviewers agree that the paper is simple and achieves impressive results. R1 found the experimental section to be thorough. R2 and R3 mention that both future and past words should be attended to. R4 presents some prior work that should be incorporated into the paper.